

# Enhanced trajectory tracking for quadrotors: disturbance observer state feedback control

Siyu Ren, Liuping Wang and Robin Ping Guan

School of Engineering, RMIT University, Melbourne, Victoria, Australia

## ABSTRACT

This study investigates the dynamics and trajectory tracking of quadcopters by utilizing the Disturbance Observer-based Control (DOBC) algorithm. The quadcopter's dynamic model, which consists of six degrees of freedom, includes both disturbances and uncertainties in the model. The DOBC approach incorporates the disturbance model into the system by introducing it at the input variables. It then compensates for disturbances to achieve accurate tracking of different reference trajectories. The simulated trajectories span a range of motion, varying from simple straight paths to complex spiral paths. In order to verify and evaluate the efficacy of the suggested control technique, simulations are performed using MATLAB. The simulations conclusively show that the disturbance observer-based method effectively achieves the tracking of specified reference trajectories in three-dimensional space. The study highlights the effectiveness of the DOBC algorithm in reducing the effects of disturbances and uncertainties, thereby improving the quadcopter's capacity to accurately track various trajectories.

## INTRODUCTION

The widespread use of quadcopters in various industries highlights their ability to be used in different applications, including important roles in military and law enforcement operations as well as essential functions in fire control departments (*Jha, 2017*; *Sebbane, 2018*; *Castiglioni et al., 2017*). Quadcopters are strategically deployed in various sectors and have a crucial role in revolutionizing agriculture by offering valuable monitoring capabilities. Furthermore, their proficiency in quickly capturing incidents or events has established them as essential instruments in the field of electronic media coverage (*Sriram Reddy & Nippun Kumaar, 2021*).

Manufacturers are producing a large number of quadcopters in response to the increasing demand for these aerial vehicles (*Deepak & Singh, 2016*). As production goes up, it becomes clearer how important it is to develop new algorithms that can accurately track references and make it easier for quadcopters to ignore disturbances (*Mendoza-Soto, Corona-Sánchez & Rodríguez-Cortés, 2018*; *Song, Zhao & Theil, 2023*; *Garlick & Bradley, 2021*). The need to increase the effectiveness of quadcopters in various fields is what drives the primary

Corresponding author
Siyu Ren, siyuren_academic@126.com

emphasis on improving algorithms in this constantly changing environment. The pursuit of resilient algorithms is in line with the primary objective of fully exploiting the capabilities of quadcopter technology and guaranteeing its smooth incorporation into various operational settings.

The distinctive features of rotorcrafts and helicopters, including their proficiency in vertical takeoff and landing combined with exceptional maneuverability, make them essential in both manned and unmanned aerial vehicle applications (*García Carrillo et al., 2013*). Significantly, in the field of unmanned aerial vehicles (UAVs), their forceful movements caused by a small moment of inertia increase the difficulty of determining their current state and require accurate implementation of controllers (*Ji, Ma & Ge, 2020*; *Dalwadi, Deb & Ozana, 2023*; *Mahony, Kumar & Corke, 2012*; *Derafa, Madani & Benallegue, 2006*).

The quadcopter, a sophisticated device that combines electronics, mechanics, and the fundamental principles of aviation, represents this intricate nature (*Chen et al., 2016*). Therefore, the development of control algorithms requires a comprehensive approach that takes into account multiple factors. It is important to prioritize the resolution of input and output disturbances (*Ma et al., 2023*) while ensuring the implementation of effective and efficient control actions (*Cole & Wickenheiser, 2019*; *Wang & Guan, 2022*; *Xuan-Mung et al., 2022*). This comprehensive strategy considers the complex interaction among the device's mechanical structure, electronic systems, and the dynamic flight environment it operates in.

The design of an efficient control algorithm is crucial to successfully navigating the complex details of quadcopters, with the ultimate objective of maximizing their performance in various operational situations (*Li, Ge & Lee, 2021*; *Zhang, Zhou & Li, 2023*; *Zhang et al., 2023*). This approach guarantees both stability and precision during flight and highlights the ongoing development of control methods to meet the changing requirements of unmanned aerial vehicle technology (*Shi et al., 2023*; *Chen et al., 2022*; *Lu et al., 2022*).

Moreover, within the wide range of research studies focused on quadcopters, a crucial area of investigation is the analysis of how disturbances affect the state inputs of these aerial vehicles (*Islam, Liu & El Saddik, 2015*). This factor has a substantial impact on the simulation results in terms of their flight paths. This study investigates the utilization of a disturbance observer-based control method (*Li et al., 2014*; *Chen et al., 2016*; *Wang & Guan, 2022*; *Liu, Chen & Shi, 2022*) to effectively address this crucial factor. The quadcopter's behavior is mathematically described by a set of nonlinear equations (*Joos et al., 2017*; *Huang, 2022*), which are sometimes approximated as linear equations when certain assumptions are made.

The system model in quadcopter position control accounts for disturbances that affect the state inputs, utilizing closed-loop feedback control. From the perspective of control system design, it is particularly effective to develop a state feedback control system that includes input disturbance estimation (*Wang & Guan, 2022*). A comparison of simulation results using various disturbance models demonstrates the superiority of simulation outcomes when the disturbance model is in line with the scenario. This methodology

exhibits impressive tracking abilities that are well-suited for different trajectory needs, indicating significant potential in the field of quadcopter aviation.

## QUADCOPTER MODEL

### Quadcopter dynamics

In quadrotor control, the body frame allows for the definition of the total thrust and rotational torques asthe total thrust T and rotational torques in the body frame can be defined as

$$
\begin{aligned}
\mathrm{T}_{total} &= \sum_{i=1}^{4} T_i = b_t \sum_{i=1}^{4} \omega_i^2, \\
\tau_\phi &= d_{mm} b_t \left(\omega_4^2 - \omega_2^2\right), \\
\tau_\theta &= d_{mm} b_t \left(\omega_3^2 - \omega_1^2\right), \\
\tau_\psi &= k_d \left(\omega_1^2 + \omega_3^2 - \omega_2^2 - \omega_4^2\right),
\end{aligned}
\tag{1}
$$

where $k_d$ is the drag constant, $\omega_i$ is the angular speed of the $i_{th}$ rotor, $d_m m$ denotes the distance extending from the rotor to the center of mass of the quadcopter, $b_t$ represents the thrust constant, which is dependent on variables including air density, as well as the dimensions of the quadcopter propeller, specifically its length and radius.

The rotor torque $\tau_{Motor_i}$ around its axis of rotation is opposite to the aerodynamic drag, which leads to

$$
\tau_{M_i} = b \omega_i^2 + I_M \dot{\omega}_i,
\tag{2}
$$

where $b$ is also the drag constant and $I_M$ is the inertia moment associated with the quadcopter's rotor. The $\dot{\omega}$ can be neglected under quasi-static state, so that

$$
\tau_{M_i} = b \omega_i^2.
\tag{3}
$$

For each rotor, the rotation axis also moves in the fixed frame of the fuselage, thus generating the gyroscopic torque

$$
\tau_{gyro} = \sum_{i=1}^{4} I_M \left(\Omega \times U_z\right) \omega_i,
\tag{4}
$$

where $U_z$ is the unit vector along inertial $z$-axis and $\Omega$ is the angular velocity.

When transitioning from the quadcopter's body coordinate system to the inertial coordinate system, a rotation matrix is employed below,

$$
\mathrm{R}_o = \begin{bmatrix} C_\theta C_\psi & S_\theta S_\phi C_\psi - C_\phi S_\psi & S_\theta C_\psi C_\phi + S_\phi S_\psi \\ C_\theta S_\psi & S_\theta S_\phi S_\psi + C_\phi C_\psi & S_\theta S_\psi S_\phi - S_\phi C_\psi \\ -S_\theta & S_\phi C_\theta & C_\phi C_\theta \end{bmatrix},
$$

where $S_x = \sin(x)$ and $C_x = \cos(x)$, with $x = \theta$ or $\psi$. In addition, from the angular velocities $\dot{\eta} = [\dot{\phi}, \dot{\theta}, \dot{\psi}]^T$ of inertial frame to the rates of angular motion $\Omega$, where $\Omega = [p, q, r]^T$, we have

$$
\Omega = \boldsymbol{F} \dot{\eta},
\tag{5}
$$

where the transformation matrix is

$$F = \begin{bmatrix} 1 & 0 & -\sin\theta \\ 0 & \cos\phi & \sin\phi\cos\theta \\ 0 & -\sin\phi & \cos\phi\cos\theta \end{bmatrix}. \tag{6}$$

For the translational dynamics of the quadcopter establishing. According to Newton's second law,

$$m \begin{bmatrix} \ddot{x} \\ \ddot{y} \\ \ddot{z} \end{bmatrix} = \begin{bmatrix} 0 \\ 0 \\ -mg \end{bmatrix} + R_o \begin{bmatrix} 0 \\ 0 \\ T_{total} \end{bmatrix}. \tag{7}$$

Expending the equation, we have

$$\ddot{x} = -\frac{k_x}{m}\dot{x} + \frac{T_{total}}{m}(\sin\theta\cos\psi\cos\phi + \sin\phi\sin\psi),$$
$$\ddot{y} = -\frac{k_y}{m}\dot{y} + \frac{T_{total}}{m}(\sin\theta\sin\psi\cos\phi - \sin\phi\cos\psi), \tag{8}$$
$$\ddot{z} = -\frac{k_z}{m}\dot{z} + \frac{T_{total}}{m}\cos\phi\cos\theta - g,$$

where $\ddot{x}$, $\ddot{y}$, and $\ddot{z}$ represent the acceleration in their respective axes. $kx$, $ky$ and $kz$ denote the coefficients of air friction, while $\dot{x}, \dot{y}, and \dot{z}$ are the velocities in the $x-$, $y-$, and $z-$ axes, respectively.

In accordance with Euler's equation of motion, the rotational dynamics of quadcopter is

$$\begin{bmatrix} \dot{p} \\ \dot{q} \\ \dot{r} \end{bmatrix} = \begin{bmatrix} 1/I_x & 0 & 0 \\ 0 & 1/I_y & 0 \\ 0 & 0 & 1/I_z \end{bmatrix} \begin{bmatrix} \tau_\phi \\ \tau_\theta \\ \tau_\psi \end{bmatrix} + \begin{bmatrix} qr(I_y - I_z)/I_x \\ pr(I_z - I_x)/I_y \\ pq(I_x - I_y)/I_z \end{bmatrix}.$$

Simplify the above equation, which leads to

$$\dot{p} = \frac{\tau_\phi}{I_x} + \frac{rq(I_y - I_z)}{I_x},$$
$$\dot{q} = \frac{\tau_\theta}{I_y} + \frac{rp(I_z - I_x)}{I_y}, \tag{9}$$
$$\dot{r} = \frac{\tau_\psi}{I_z} + \frac{qp(I_x - I_y)}{I_z},$$

where $\dot{p}$, $\dot{q}$, and $\dot{r}$ stand for the angular acceleration parameters within the quadcopter body-centric coordinate system. $I_x$, $I_y$ and $I_z$ represent the inertia tensors associated with the $x-$, $y-$, and $z-$ axes, respectively.

Moreover, the relationship between the Euler angular velocities $\dot{\phi}, \dot{\theta}, \dot{\psi}$ and the angular velocities $(p, q, r)$ is expressed as follows

$$\begin{bmatrix} \dot{\phi} \\ \dot{\theta} \\ \dot{\psi} \end{bmatrix} = \begin{bmatrix} 1 & T_\theta S_\phi & T_\theta C_\phi \\ 0 & C_\phi & -S_\phi \\ 0 & \frac{S_\phi}{C_\theta} & \frac{C_\phi}{C_\theta} \end{bmatrix} \begin{bmatrix} p \\ q \\ r \end{bmatrix}. \tag{10}$$

Expending the equation, we have

$$
\dot{\phi} = p + qT_\theta S_\phi + rT_\theta C_\phi,
$$
$$
\dot{\theta} = qC_\phi - rS_\phi, \tag{11}
$$
$$
\dot{\psi} = q\frac{S_\phi}{C_\theta} + r\frac{C_\phi}{C_\theta},
$$

where $T_x = tan(x)$, $S_x = sin(x)$, and $C_x = cos(x)$, with $x = \phi$ or $\theta$; $\dot{\phi}, \dot{\theta}$, and $\dot{\psi}$ are the angular velocities within the inertial coordinate system along the $x-$, $y-$, and $z-$ axes, respectively.

## State space model

Define $\dot{x} = a$, $\dot{y} = b$ and $\dot{z} = c$, the state space model contains 12 state variables for the highly coupled quadcopter system:

$$
\dot{x} = a,
$$
$$
\dot{y} = b,
$$
$$
\dot{z} = c,
$$
$$
\dot{a} = -\frac{k_x}{m}a + \frac{T_{total}}{m}(\sin\theta\cos\psi\cos\phi + \sin\phi\sin\psi),
$$
$$
\dot{b} = -\frac{k_y}{m}b + \frac{T_{total}}{m}(\sin\theta\sin\psi\cos\phi - \sin\phi\cos\psi),
$$
$$
\dot{c} = -\frac{k_z}{m}c + \frac{T_{total}}{m}\cos\phi\cos\theta - g,
$$
$$
\dot{p} = \frac{\tau_\phi}{I_x} + \frac{rq(I_y - I_z)}{I_x},
$$
$$
\dot{q} = \frac{\tau_\theta}{I_y} + \frac{rp(I_z - I_x)}{I_y},
$$
$$
\dot{r} = \frac{\tau_\psi}{I_z} + \frac{qp(I_x - I_y)}{I_z},
$$
$$
\dot{\phi} = p + qT_\theta S_\phi + rT_\theta C_\phi,
$$
$$
\dot{\theta} = qC_\phi - rS_\phi,
$$
$$
\dot{\psi} = q\frac{S_\phi}{C_\theta} + r\frac{C_\phi}{C_\theta}.
$$

From the above equations, we have

$$
\dot{X} = f(X, U),
$$

where $X = (x, y, z, a, b, c, p, q, r, \phi, \theta, \psi)^T$ represents the state vertor, $U = (T_{total}, \tau_\phi, \tau_\theta, \tau_\psi)$ denotes the thrust vector acting as control input, and $Y = (x, y, z, \theta)$ is the output vector. To ensure precise trajectory tracking, the initial focus lies on confirming the alignment of the first three vectors concerning position control. Given the interconnected nature of the quadcopter, an additional check involves validating an angular velocity along the $y-$axis. This acts as an indirect measure to ascertain the accuracy of the quadcopter's interlinked system.

# DISTURBANCE OBSERVER DESIGN

The disturbance observer (DOB) is an advanced control approach used in engineering to accurately estimate and effectively prevent unexpected interruptions that occur within control systems. The main purpose of the system is to measure disruptions that arise from several sources, including external environmental factors, oscillations within the system, or errors in the modeling process. These disruptions provide a potential danger to the performance of the control system and require proactive measures to be taken through the observer's assessments.

The fundamental principle of DOB is typically based on the mathematical depiction of the system, encompassing its state variables and outputs. By conducting a comparison between the projections of the model and the actual behavior of the system, DOB is able to detect inconsistencies, which are frequently caused by disturbance signals. By evaluating these disruptions, DOB produces a corrective signal that is incorporated into the input of the control system. This effectively reduces the negative effects of the disturbances and maintains the desired performance and stability of the system.

## Quadcopter input disturbance observer design

Initially, it presupposes an input disturbance denoted as $v(i)$ in the system, which subsequently results in a transformation of the state-space model.

$$x_n(i+1) = A_n x_n(i) + B_n(u(i) + v(i)),$$

$$y(i) = C_n x_n(i), \tag{12}$$

where $u(i)$ and $v(i)$ stand for the control inputs, $y(i)$ represents the system responses, and $x_n(i)$ represents the state variable set.

The input disturbance $v(i)$ is conceptualized as

$$v(i) = \frac{j^{-1}\epsilon(i)}{D(j^{-1})}, \tag{13}$$

where $\epsilon(i)$ is consistent with the dimension of vector $v(i)$.

Let the disturbance model $D(j^{-1})$ be

$$D(j^{-1}) = 1 + d_1 j^{-1} + d_2 j^{-2} + d_3 j^{-3} + \ldots + d_\gamma j^{-\gamma}. \tag{14}$$

Next, use the form of difference equation to express the disturbance vector $v(i+1)$:

$$v(i+1) = -d_1 v(i) - d_2 v(i-1) - \ldots - d_\gamma v(i-\gamma+1) + \epsilon(i), \tag{15}$$

where $\epsilon(i)$ represents white noise with a mean value of zero.

Following our previous study (*Wang & Guan, 2022*), by introducing matrix $h(i)$ of size $n\gamma \times 1$ to represent the disturbance group and presenting it with a state space mode, we have

$$h(i) = \begin{bmatrix} v^T(i) & v^T(i-1) & \cdots & v^T(i-\gamma+1) \end{bmatrix}^T. \tag{16}$$

Then, using the state space model below to obtain the disturbance

$$h(i+1) = A_{dis}h(i) + B_\epsilon \epsilon(i),$$

$$v(i) = C_\epsilon h(i). \tag{17}$$

For matrix $A_{dis}$, it shows the form as below:

$$A_{dis} = \begin{bmatrix} -d_1 I & -d_2 I & \dots & -d_{\gamma-1}I & -d_\gamma I \\ I & O & \dots & \dots & O \\ O & I & O & \dots & \vdots \\ \vdots & \ddots & \ddots & \ddots & \vdots \\ O & \dots & O & I & O \end{bmatrix},$$

where $B_\epsilon$ is a $n\gamma \times n$ matrix and $C_\epsilon$ is a $n \times n\gamma$ matrix, they both have the $n \times n$ identity matrix and the rest vertical and horizontal elements of the matrix are zeros. The leftover rows and columns in both matrices consist of zero matrices. $O$ is a $n \times n$ zero matrix and $I$ is the identity matrices with same dimensions.

For the disturbance $D(j^{-1})$, we have

$$det\left(j^{-1}I - A_d\right) = D(j^{-1})^n, \tag{18}$$

where $D(j^{-1})$ is the disturbance model in backward shift form. Matrix $A_{dis}$ includes $n$ groups of eigenvalues which related to the zeros of the $D(j^{-1})$.

The incorporation of the disturbance vector $v(i)$ into the state space model serves to mitigate the impact of external interference on the state estimate matrix $x_n(i)$. The introduced augmented state matrix is

$$x(i) = \left[ x_n^T(i) \quad h(i)^T \right]^T.$$

We rewrite the previous state space model as

$$x_n(i+1) = A_n x_n(i) + B_n C_\epsilon h(i) + B_n u(i).$$

Under this condition, an augmented system space model is derived:

$$\begin{bmatrix} x_n(i+1) \\ h(i+1) \end{bmatrix} = \overbrace{\begin{bmatrix} A_n & B_n C_\epsilon \\ O_1 & A_{dis} \end{bmatrix}}^{\mathcal{A}} \begin{bmatrix} x_n(i) \\ h(i) \end{bmatrix}$$

$$+ \overbrace{\begin{bmatrix} B_n \\ O_2 \end{bmatrix}}^{B} u(i) + \begin{bmatrix} O_3 \\ B_\epsilon \end{bmatrix} \epsilon(i),$$

where $O_{1,2,3}$ are zero matrices to adapt to corresponding dimensions; the integrated $A$ and $B$ matrices are regarded as the augmented matrices of the system.

The output is computed as

$$y(i) = \overbrace{\begin{bmatrix} C_n & O_4 \end{bmatrix}}^{C} \begin{bmatrix} x_n(i) \\ h(i) \end{bmatrix},$$

where $C$ is the augmented output matrix, and $O_4$ is a zero matrix like $O_{1,2,3}$.

 

After establishing the augmented model, we will design the observer on this basis, $A$ and $C$ matrices will be used in the design of this step. First define an observer gain $K_{ob}$, it is a matrix with the same dimension as $A$ and $C$. To get the value of $K_{ob}$, we use the discrete-time LQR algorithm. The selected observer gain is to stabilize the state matrix $(A - K_{ob}C)$ in the closed-loop observer error system deviation dynamics. As a result, the augmented state matrix $x(i)$ becomes amenable to estimation.

$$\hat{x}(i+1) = A\hat{x}(i) + Bu(i) + K_{ob}(y(i) - C\hat{x}(i)). \tag{19}$$

Noting that in order to ensure that $A$ and $C$ are observable, it is first necessary to ensure that the $An$ and $Cn$ matrices are observable, and the system has no zeroes which related to $D(j^{-1})$. Under this condition, the observer can operate effectively.

## Disturbance observer-based discrete-time state feedback control

Before appling augmented system state representation, converting continuous model to discrete model. The state space model in continuous time can be formulated as follows:

$$\dot{x}_n(k) = A_c x_n(k) + B_c u(k), \tag{20}$$

$$y(k) = C_c x_n(k). \tag{21}$$

The discrete-time formulation of the state space model can be succinctly articulated as follows:

$$x_n(i+1) = A_n x_n(i) + B_n u(i), \tag{22}$$

$$y(i) = C_n x_n(i). \tag{23}$$

Based on the design of adding disturbance to the input and its compensation in the state estimation, the design of the state feedback control is undertaken through the subsequent stepwise procedure. It continues to adopt the state-space formulation detailed in the preceding section, following the design of model:

$$x_n(i+1) = A_n x_n(i) + B_n(u(i) + v(i)),$$
$$y(i) = C_n x_n(i).$$

Noting that the input vectors are $u(i)$ and $v(i)$. It assumes that the observational properties are satisfied for matrices $An$ and $Cn$ and that the controllability conditions are met for the pair $(A_n, B_n)$. Moreover, the system avoid the zeros which related to the disturbance occurring at the $D(j^{-1})$.

We define the intermediary control signal at this stage:

$$\tilde{u}(k) = u(k) + \mu(k).$$

Then the original $x_n(i+1)$ becomes

$$x_n(i+1) = A_n x_n(i) + B_n \tilde{u}(i). \tag{24}$$

Following our previous study (*Wang & Guan, 2022*), the principle behind state feedback control using a disturbance observer involves designing the state feedback controller $K$ specifically for the intermediate control signal $\tilde{u}(k)$. This controller aims to handle the estimated variable to effectively compensate for the input disturbance $v(k)$.

It integrates $(x_n(i+1), h(i+1))$ and $y(i)$ as a complete augmented model

$$
\begin{bmatrix} x_n(i+1) \\ h(i+1) \end{bmatrix} = \overbrace{\begin{bmatrix} A_n & B_n C_\epsilon \\ O_1 & A_{dis} \end{bmatrix}}^{A} \begin{bmatrix} x_n(i) \\ h(i) \end{bmatrix}
$$

$$
+ \overbrace{\begin{bmatrix} B_n \\ O_2 \end{bmatrix}}^{B} u(i) + \begin{bmatrix} O_3 \\ B_\epsilon \end{bmatrix} \epsilon(i)
$$

$$
y(i) \qquad = \overbrace{\begin{bmatrix} C_n & O_4 \end{bmatrix}}^{C} \begin{bmatrix} x_n(i) \\ h(i) \end{bmatrix}.
$$

Based on this model, we can use the disturbance compensation strategy to get $\hat{x}_n(i)$ and $\hat{v}(i)$ from previous disturbance observer design section.

Deriving the controller $K$ utilizing matrices $A_n$ and $B_n$ to ensure the stability of the closed-loop control system matrix $A_n - B_n K$, the interim control variable $\tilde{u}(i)$ is computed as

$$\tilde{u}(i) = -K\hat{x}_n(i). \tag{25}$$

Secondly, get the observer gain $K_{ob}$ using matrices $A$ and $C$ allowing for the estimation of both the input disturbance $v(i)$ and the state variable $x_n(k)$ within the augmented state space model, resulting in $\hat{v}(i)$ and $\hat{x}_n(i)$.

After that, deducting $\hat{v}(i)$ from $\tilde{u}(i)$ results in the control variable $u(i)$:

$$u(i) = \tilde{u}(i) - \hat{v}(i). \tag{26}$$

Compare a fixed signal $r(i)$ with our output signal, considering the reference signal $r(i)$, where the observer formulation takes the following form

$$\hat{x}(i+1) = A\hat{x}(i) + Bu(i) + K_{ob}(y(i) - r(i) - C\hat{x}(i)).$$

The $(A, B, C)$ matrices are augmented matrices which contains the input disturbance compensation. Furthermore, it is imperative to validate that the entirety of the eigenvalues pertaining to the observer error system matrix $A - K_{ob}C$ remains strictly bounded within the confines of the unit circle, thereby guaranteeing system stability and observability.

Determined the closed-loop eigenvalue of the state feedback control system based on disturbance observer to ensure the stability of the system. For both $x_n(i)$ and $\hat{p}(i)$, we adopt the observation error system

$$
\begin{bmatrix} E_{xp}\{\tilde{x}_n(i+1)\} \\ E_{xp}\{\bar{h}(i+1)\} \end{bmatrix} = (A - K_{ob}C) \begin{bmatrix} E_{xp}\{\tilde{x}_n(i)\} \\ E_{xp}\{\bar{h}(i)\} \end{bmatrix},
$$

where $E_{xp}$ represents the error variables $x_n(i)$ and $\hat{h}(i)$, with their expectations denoted as respective values.

In addition, the proposed disturbance observer state feedback control strategy is compared to the traditional Proportional-Integral-Derivative (PID) control strategy based on important performance indicators, including trajectory tracking accuracy, response time, resilience to external disturbances, and algorithm complexity. At first, the proposed technique demonstrates superior precision in accurately following a desired path compared to the classic PID control, especially when dealing with external disturbances. Furthermore, it demonstrates greater proficiency in maneuvering intricate and ever-changing surroundings. In terms of response time, this technique efficiently adjusts control parameters to accommodate variations in external variables, surpassing the performance of conventional PID control. Our system, which includes the use of a disturbance observer, is highly effective in resisting external disturbances. It is particularly adept at withstanding environmental changes like strong winds, ensuring excellent flight stability and trajectory accuracy. This sets it apart from conventional methods. Furthermore, although the complexity of our method's control algorithm exceeds that of PID control, its benefits in terms of other performance metrics justify this complexity.

## EXPERIMENTS

For simulation, the mass of the UAV is set to 0.5 kilograms, The scale factor of gravity is $g = 9.80122$ N/kg. The system model is linearized using the Taylor series expansion approach, specifically targeting equilibrium points and utilizing the Jacobian linearization methodology for simulation purposes. The sampling instant is chosen to 900.

In the context of state feedback control utilizing disturbance observers, we choose a specific polynomial $D(j^{-1})$ as

$$D(j^{-1}) = 1 - 2\cos \omega_d j^{-1} + j^{-2},$$

where $\omega_d = 0.5$. When the aerial reference trajectory follows a linear path, the term $\cos\omega_d$ will be removed.

We assume the quadcopter system has the input disturbance on each axis, for $x-$ and $y-$axes, the input disturbance $\mu(k)$ can be expressed as

$$v(i) = \frac{\epsilon(i)}{\left(1 - 2\cos \omega_d j^{-1} + j^{-2}\right)}. \tag{27}$$

This is the main disturbance signal that we test in this paper. In addition, if the controller is an integrator type, it chooses step or random walk type disturbance signal $1 - q^{-1}$ as $D(q^{-1})$, if the controller is an integrator plus sinusoidal mode, $D(j^{-1})$ will also be changed to

$$(1 - j^{-1})(1 - 2\cos \omega_d j^{-1} + j^{-2}).$$

According to the augmented state space equation, it can accomplish the state feedback control system with an estimation of the input disturbance.

Initially, it obtains the augmented state space model $(A, B, C)$ with $(An, Bn, Cn)$. Since we have 12 state variables, four input variables and four outputs for quadcopter modeling at the begining. $(An, Bn, Cn)$ are $12 \times 12$, $12 \times 4$ and $4 \times 12$ matrices respectively. The dimensions of $A$, $B$, and $C$ are $20 \times 20$, $20 \times 4$ and $4 \times 20$, respectively.

Utilizing the MATLAB function `dlqr.m` with $Q$ as the identity matrix and $R$ set to one, the state feedback controller is systematically designed based on the pair $(An, Bn)$, yielding

$$
K = \begin{bmatrix}
0 & 0 & 0.3172 & 0 \\
0 & -0.3172 & 0 & 0 \\
0.9766 & 0 & 0 & 0 \\
0 & 0 & 0.3841 & 0 \\
0 & -0.3841 & 0 & 0 \\
1.1623 & 0 & 0 & 0 \\
0 & 1.6566 & 0 & 0 \\
0 & 0 & 1.6566 & 0 \\
0 & 0 & 0 & 0.4875 \\
0 & 0.3536 & 0 & 0 \\
0 & 0 & 0.3536 & 0 \\
0 & 0 & 0 & 0.5004
\end{bmatrix}^{T}.
$$

This state feedback regulator $K$ is used for manipulating the interim control parameter $\tilde{u}(i)$, thereby offsetting the external disturbance $v(i)$ using the estimated parameter $\hat{v}(i)$. In addition, ensure that the matrix $A_n - B_n K$ lies within the unit circle, thereby maintaining the stability of the closed-loop control system. Computing the intermediary control variable $\tilde{u}(i)$ based on the previous equation:

$$
\tilde{u}(i) = -K\hat{x}_n(i). \tag{28}
$$

With the augmented state space model strategy using the disturbance model, derive the observer gain $K_{ob}$ utilizing the augmented matrices $A$ and $C$, followed by estimating both the input disturbance $v(i)$ and the state variable $x_n(i)$, resulting $\hat{v}(i)$ and $\hat{x}_n(i)$, where the

$K_{ob}$ equals to

$$
\begin{bmatrix}
1.2350 & 0 & 0 & 0 \\
0 & 1.2350 & 0 & 0 \\
0 & 0 & 0.8012 & 0 \\
25.0089 & 0 & 0 & 0 \\
0 & 25.0089 & 0 & 0 \\
0 & 0 & 6.2022 & 0 \\
0 & -33.2883 & 0 & 0 \\
33.2883 & 0 & 0 & 0 \\
0 & 0 & 0 & 1.3037 \\
0 & -275.4516 & 0 & 0 \\
275.4516 & 0 & 0 & 0 \\
0 & 0 & 0 & 28.1125 \\
0 & 0 & 14.1530 & 0 \\
0 & -10.4817 & 0 & 0 \\
10.4817 & -0 & 0 & 0 \\
0 & 0 & 0 & 4.9595 \\
0 & 0 & 13.7672 & 0 \\
0 & -10.0984 & 0 & 0 \\
10.0984 & 0 & 0 & 0 \\
0 & 0 & 0 & 4.3870
\end{bmatrix}.
$$

Once got the intermediary control signal $\tilde{u}(i)$ and estimated input $\hat{v}(i)$, the previous difference equation is used to obtain $u(i)$

$$u(i) = \tilde{u}(i) - \hat{v}(i). \tag{29}$$

Comparing the output vector $y(i)$ with the target signal $r(i)$, the observer equation can be written as

$$\hat{x}(i+1) = A\hat{x}(i) + Bu(i) + K_{ob}(y(i) - r(i) - C\hat{x}(i)).$$

A sensitivity analysis is performed to evaluate the effectiveness of the control algorithm under different parameter settings. This is achieved by modifying various parameters of the quadcopter unmanned aerial vehicle, such as mass, moment of inertia, and aerodynamic properties, within a simulated environment. This study includes a wide range of parameter adjustments, including the mass moment of inertia.

The control approach extensively depends on the Dryden turbulence model to accurately measure the impact of wind disturbances. The Power Spectral Density atmospheric turbulence model represents wind disturbances as stochastic fluctuations in a continuous random process. White noise, which has a consistent power spectral density and equal strength across certain frequency ranges, is subjected to shaping filtration in order to create the intended turbulence effects.

Disturbances are intentionally produced in a predetermined direction for a quadcopter drone operating in space in our simulated scenario. The transfer functions of the filter are

expressed as follows:

$$H_u(s) = \sigma_u \sqrt{\frac{2L_u}{\pi V}} \frac{1}{1 + \frac{L_u}{V}s}, \tag{30}$$

where $L_u$ and $\sigma_u$ represent the scale length of turbulence and turbulence intensity, respectively; $V$ represents the air velocity.

Once the white noise is filtered, it is included into the simulation model, and the resulting simulation results are thoroughly evaluated.

## RESULTS

### Trajectory types

First of all, we have assumed three cases for the reference trajectory in 3D space. The first case is a single straight-line trajectory. The second case is the cylindrical spiral track, which contains sine and cosine curves and can represent the turning or hovering situation of the UAV in general. The third case is the second conic trajectory, which can be understood as a high-order extension in the second case. Because the conic curve contains elliptical, parabolic, and hyperbolic trajectories, it can represent more realistic UAV flight trajectories and has certain representativeness in the aviation field. The simulation results are presented below.

**Reference following.** Figure 1 shows the reference tracking performance with the integrator mode controller and random walk disturbances: $D(j^{-1}) = 1 - j^{-1}$.

Figure 2 illustrates the simulation results with a sinusoidal mode controller and the disturbance in the corresponding sinusoidal mode: $D(j^{-1}) = 1 - 2\cos \omega_d j^{-1} + j^{-2}$.

Figure 3A shows that the reference signal is extended to a conic track with a variable turning radius. In addition, Fig. 3B shows an intuitive 3D space simulation result. Figure 1 to Fig. 3 show that for different reference trajectories and controllers, selecting appropriate corresponding disturbance models $D(j^{-1})$ can complete the disturbance compensation issue and make the simulation results consistent with the reference trajectories.

To experiment with a contrast test on the premise of conic trajectory, if the $D(j^{-1})$ model is changed to not correspond to the controller, the quadcopter will not be able to effectively complete the compensation of disturbance and trajectory tracking. The simulation results are shown in Fig. 4, taking the $y-$axis as an example.

Unmanned aerial vehicles may undergo alterations in their mass and moment of inertia during actual use, which can be caused by factors such as cargo variations, battery usage, or ambient conditions. Therefore, it is essential to evaluate the stability and efficiency of the control strategy in light of these variations in parameters. This study thoroughly examines the resilience of the suggested control approach when faced with different aircraft parameter fluctuations.

The analysis considers the fluctuation in the weight of the unmanned aerial vehicle, utilizing simulated scenarios that incorporate a gradual rise in weight up to 1.5 kg. Our simulation results show that the suggested control method maintains a high level of trajectory tracking performance even when there are changes in mass that affect how the

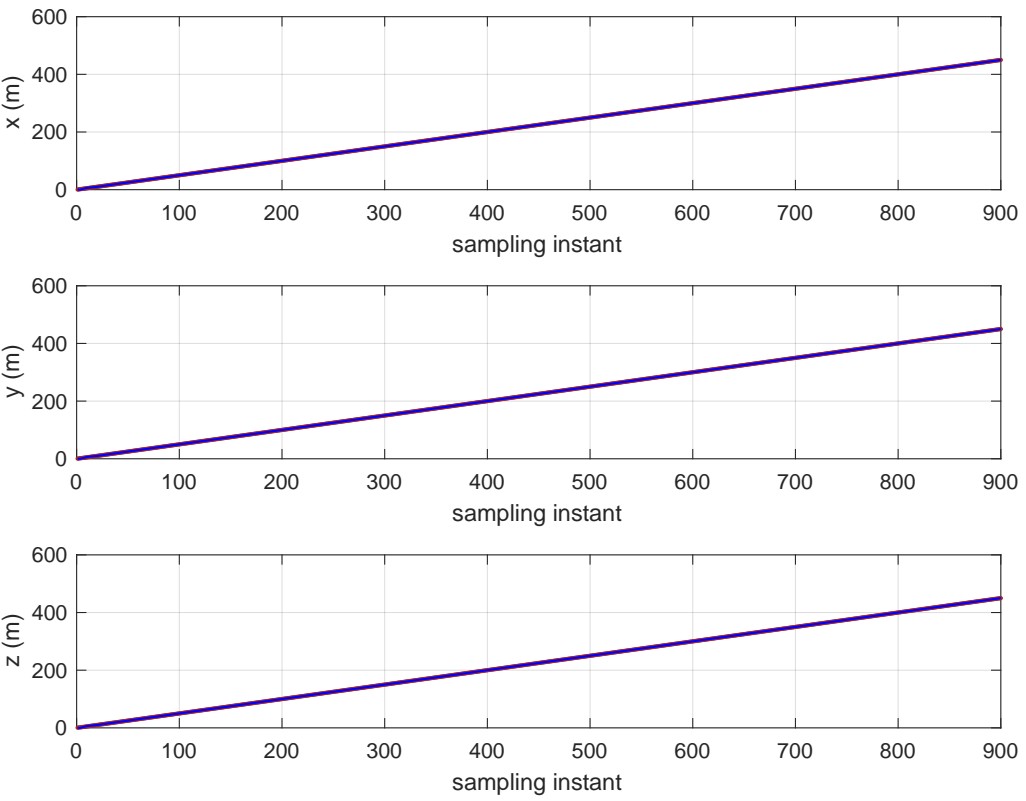

**Figure 1** Reference signal (brown) and following results (blue) with a steep slope track in the x-, y-, and z-axes.

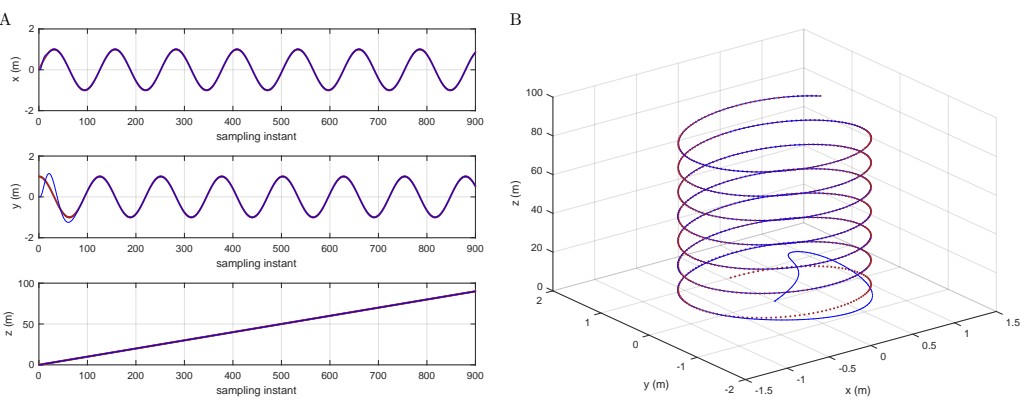

**Figure 2** Reference signal (brown) and following results (blue) with a cylindrical spiral track (A) in the x-, y-, and z-axes and (B) in 3D space.

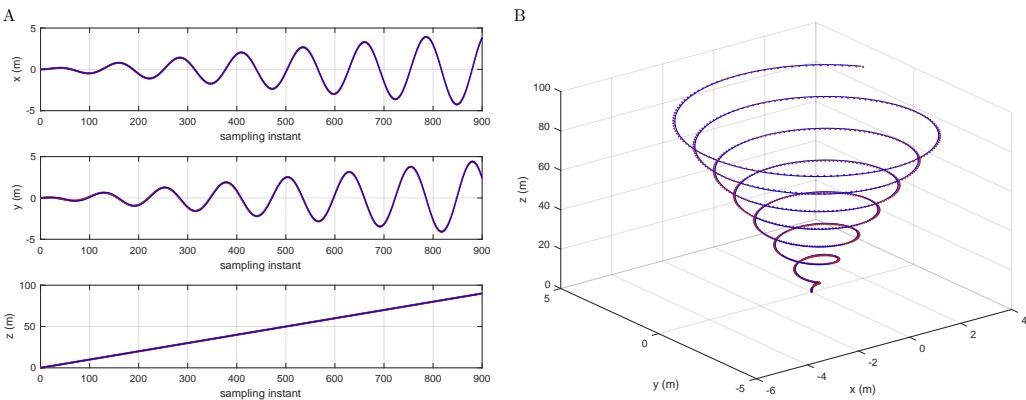

**Figure 3** Reference signal (brown) and following results (blue) with a conic track (A) in the x-, y-, and z-axes and (B) in 3D space.

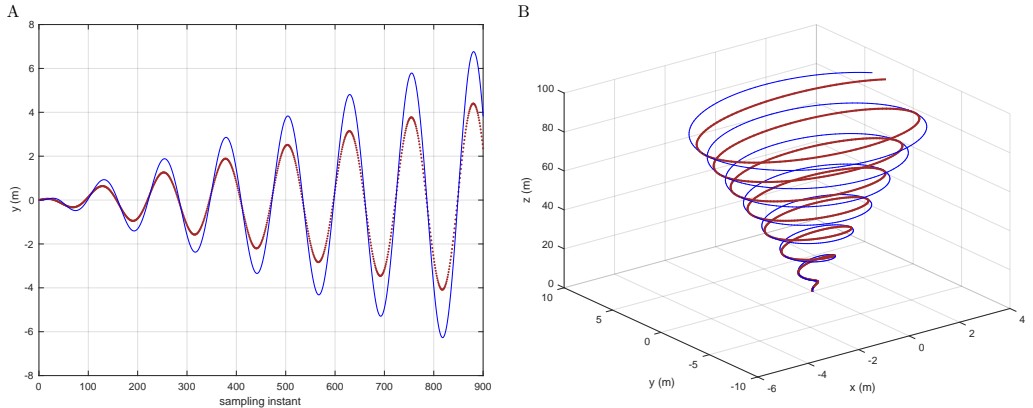

**Figure 4** Reference signal (brown) and following results (blue) based on the integral type D with conic track in the $y$-axis (A) and in 3D space (B).

system moves. This validates the efficacy of the technique in addressing fluctuations in mass. The study examines the impact of changes in the weight of the unmanned aerial vehicle, specifically focusing on cases where the weight decreases. Following a 30% decrease in mass, the trajectory tracking, as illustrated in Fig. 5A, continues to provide very good performance. The simulation effectively retains its tracking performance, similar to its performance before the drop in mass, using the $y$-axis as an illustration. It ultimately approaches the reference trajectory after approximately two sinusoidal oscillation cycles. The results of the simulation show that the suggested control method consistently achieves a high level of trajectory tracking performance, even when there are changes in mass that affect how the system moves.

Furthermore, a thorough examination of the influence of changes in the moment of inertia on the control system is conducted. After a specific decrease of 15% in inertia, the trajectory tracking, as seen in Fig. 5B, exhibits a comparable pattern. Illustrating the $y$-axis

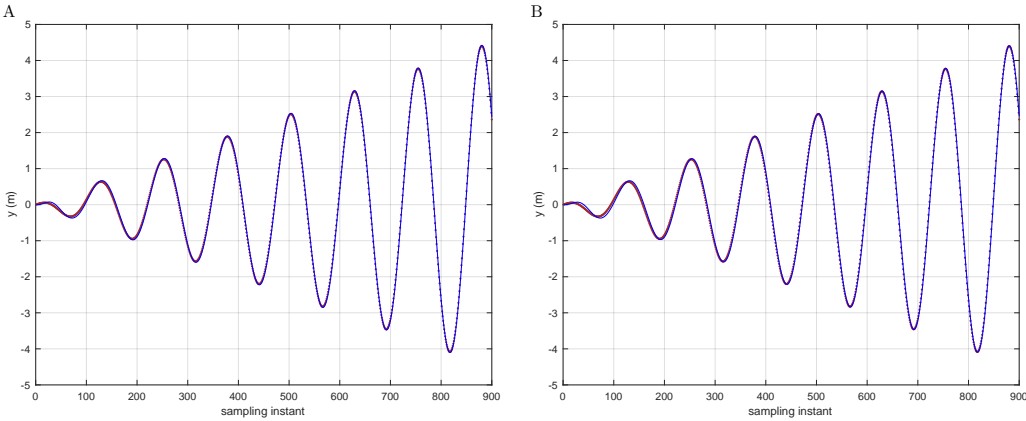

**Figure 5** Reference signal (brown) and following result (blue) with conic *y*-axis track presentation for the mass (A) and inertial (B) change conditions.

as an instance, at a small scale, the simulation presently reaches a state of convergence with the reference trajectory after around three cycles of sinusoidal oscillation. At a large scale, the general ability to accurately follow a desired path continues to be exceptional, as the error values consistently stay within the range of 0.002 m.

## CONCLUSIONS

In conclusion, this article presents a control method for tracking trajectories that utilizes disturbance-observer-based control. This method is showcased in the context of quadcopter trajectory control by simulating it across three different reference trajectories. The results demonstrate the method's ability to precisely monitor sinusoidal trajectories in three-dimensional space, with particularly impressive performance in tracking conic curves. Significantly, its flexibility encompasses intricate flight paths for UAVs, even when faced with irregular turning radii. Moreover, its potential uses range from accurate position manipulation to broader effects in controlling UAVs in aviation. This approach offers favorable opportunities in the fields of Unmanned Aerial Vehicle technology and flight control.

### Funding
The authors received no funding for this work.

### Competing Interests
The authors declare there are no competing interests.

## Author Contributions

- Siyu Ren conceived and designed the experiments, performed the experiments, analyzed the data, performed the computation work, prepared figures and/or tables, authored or reviewed drafts of the article, and approved the final draft.
- Liuping Wang conceived and designed the experiments, authored or reviewed drafts of the article, and approved the final draft.
- Robin Ping Guan conceived and designed the experiments, authored or reviewed drafts of the article, and approved the final draft.

## Data Availability

The MATLAB code and data used in the experiments are available in the Supplemental Files.

## Supplemental Information

Supplemental information for this article can be found online at http://dx.doi.org/10.7717/peerj-cs.1861#supplemental-information.

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
