# Peer review of "Enhanced trajectory tracking for quadrotors: disturbance observer state feedback control"

_PeerJ Computer Science, doi:10.7717/peerj-cs.1861_

## Round 0.1 · original submission · Major Revisions

Please revise your manuscript based on the comments from the two reviewers. Both reviewers have offered constructive feedback to enhance the manuscript. Reviewer 2, in particular, suggested the inclusion of additional experiments to further validate your findings. We strongly encourage you to consider these suggestions seriously, as they could significantly bolster the robustness and depth of your study. Should you find it challenging to conduct the suggested experiments due to constraints or limitations, we kindly request that you provide a detailed explanation in your new submission.

·

Basic reporting

+ The authors have used clear and precise language in the manuscript.
+ The technical content is clear.
+ The figures in the manuscript are of good quality. The resolution of the figures is good enough for readers to see important details. The descriptions and labels of each figure are clear and detailed.
+ The authors have presented almost all of the results that relate to their hypothesis.
+ However, the authors should describe more of the disturbance observer's design concepts and theoretical foundations, as well as how to change disturbance observer settings to accommodate diverse disturbances.
+ Also, there is some missing information about some notations in the equations that needed to be provided. Please see the additional comments below for details.

Experimental design

+ The research fits well with the Aims and Scope of the journal.
+ The experimental design is OK.
+ The authors did provide the code, which is helpful to reproduce the reported results.

Validity of the findings

+ In my opinion, this research is useful.
+ The conclusions in the manuscript are well stated and linked to the original research question.

Additional comments

+ The introduction part can be extended, for example, by including more related studies and describing the motivation for developing more robust control algorithms for quadcopters.
+ It may not be clear to the readers whether the proposed model is linearized or not.
+ What is \Omega in Equation 4?
+ Missing descriptions of some notations in Equation 8, for example, what are \ddot{x} and \dot{x}?
+ Also, in Equation 9, what are \dot{p}, \dot{q}, and \dot{r}?

Reviewer 2 ·

Basic reporting

- The authors used language that is simple to understand and relatively specific in their work.
- The main technical content can be understood for a wide range of readers.
- The figures are clear and are all in vector format. Some more figures can be included, as indicated in my comments below.
- Although some paragraphs can be extended, the manuscript includes enough background material as well as an introduction to show how the work fits into the greater field of knowledge.
- The hypothesis is supported by the experimental results. However, more experiments can be done to improve the quality of the work, as indicated below.

Experimental design

- The study question is clearly defined in the text, and the experiments sound good. However, there are a few aspects that require further explanation or experiments:
+ The author could consider performing a robustness analysis on quadcopter flight control algorithms in relation to modifications in parameters such as mass, inertia, and aerodynamic characteristics.
+ What is the type of noise in Equation (15)?
+ Environmental disturbances (e.g., wind, like those in Cole and Wickenheiser (2019) as cited in the manuscript) should be investigated in order to examine the system's performance under various noise and disturbance circumstances. The robustness of control techniques should be assessed in simulations by introducing noise with unique properties such as white noise.
- In addition, a comparison with existing control strategies should be included as well. Control precision, computational complexity, ease of implementation, and robustness could all be considered in this comparison.

Validity of the findings

- The findings presented have been verified by the results of the experiments.
- The conclusions are clear and provide an answer to the research topic.

Additional comments

- Figures 2 and 4 should include a 3D visualization of the signals, like the plot in Figure 3.
- This manuscript needs to be revised in order to address all the comments above.

---

## Round 0.2 · accepted · Accept

The authors have addressed all of the comments from the two reviewers in the previous reviewing round, and all the reviewers, including a new reviewer, support the publication of the work. The manuscript is ready for publication.

·

Basic reporting

The authors have addressed all of my concerns in this new version. I am satisfied with this version and support its publication.

Experimental design

With this updated edition, the writers have handled all of my concerns. This version meets my requirements, and I encourage it to be published.

Validity of the findings

The authors have addressed all concerns I had with the previous edition in this revised version. This version satisfies my expectations and I recommend publication.

Reviewer 2 ·

Basic reporting

Presentation and language are OK and I have no further comments.

Experimental design

The authors have done extra experiments as suggested. I have no further comments.

Validity of the findings

The findings are both valid and useful. I have no other comments.

Additional comments

The authors have conducted a significant revision and have taken into account all of my comments. In my opinion, the manuscript is now suitable for being accepted for publication.

·

Basic reporting

This paper proposes the use of a disturbance observer-based control approach for trajectory tracking in the demonstration of quadcopter trajectory control through simulation reference trajectories. The method’s ability to monitor sinusoidal trajectories in three-dimensional space is demonstrated. This flexibility for complex flight paths with irregular turning radii makes it promising UAV and flight control applications.

This is a relevant work of interest to researchers in the field of UAV flight control. The manuscript is also well-organized and written. From what I see, this version of manuscript has previously gone through rigorous review and revision. The authors have addressed the comments and suggestions brought up by the previous reviewer, which are already quite positive in the last round of review.

Experimental design

This is a relevant work of interest to researchers in the field of UAV flight control. The research question is well defined and meaningful.

Validity of the findings

The conclusion is appropriately supported by the study.